# A locomotor neural circuit persists and functions similarly in larvae and adult *Drosophila*

**Kristen Lee, Chris Q Doe***

Institute of Neuroscience, Howard Hughes Medical Institute, University of Oregon, Eugene, United States

**Abstract** Individual neurons can undergo drastic structural changes, known as neuronal remodeling or structural plasticity. One example of this is in response to hormones, such as during puberty in mammals or metamorphosis in insects. However, in each of these examples, it remains unclear whether the remodeled neuron resumes prior patterns of connectivity, and if so, whether the persistent circuits drive similar behaviors. Here, we utilize a well-characterized neural circuit in the *Drosophila* larva: the moonwalker descending neuron (MDN) circuit. We previously showed that larval MDN induces backward crawling, and synapses onto the Pair1 interneuron to inhibit forward crawling (Carreira-Rosario et al., 2018). MDN is remodeled during metamorphosis and regulates backward walking in the adult fly. We investigated whether Pair1 is remodeled during metamorphosis and functions within the MDN circuit during adulthood. We assayed morphology and molecular markers to demonstrate that Pair1 is remodeled during metamorphosis and persists in the adult fly. MDN-Pair1 connectivity is lost during early pupal stages, when both neurons are severely pruned back, but connectivity is re-established at mid-pupal stages and persist into the adult. In the adult, optogenetic activation of Pair1 resulted in arrest of forward locomotion, similar to what is observed in larvae. Thus, the MDN-Pair1 neurons are an interneuronal circuit – a pair of synaptically connected interneurons – that is re-established during metamorphosis, yet generates similar locomotor behavior at both larval and adult stages.

**\*For correspondence:**
cdoe@uoregon.edu

## Introduction

Large-scale changes in neuronal morphology and function occur during mammalian puberty (*Barendse et al., 2018*; *Mills et al., 2016*; *Sisk and Zehr, 2005*), as well as several neurobiological disorders including depression (*Patel et al., 2019*), or chronic pain (*Kuner and Flor, 2017*). Similarly, major changes in neuronal numbers and type occur as a result of insect metamorphosis (*Kanamori et al., 2015*; *Truman and Reiss, 1976*; *Yaniv and Schuldiner, 2016*). Despite these changes, there are documented cases of individual insect neurons persisting from larval to adult stages. In *Drosophila*, individual motor and sensory neurons have been shown to persist throughout metamorphosis and undergo dramatic remodeling (*Consoulas et al., 2002*; *Consoulas et al., 2000*; *Yaniv and Schuldiner, 2016*; *Yu and Schuldiner, 2014*). Similar findings have been reported for the insect mushroom body, where Kenyon cells partners (projection neurons, DANs) exist at both larval and adult stages (*Li et al., 2020*; *Marin et al., 2005*). Yet, it remains unclear whether the remodeled neurons re-establish connectivity with the identical neurons in the larva and adult.

During *Drosophila* metamorphosis, the animal changes from a crawling limbless larva to a walking six-legged adult (*Riddiford, 1980*; *Riddiford et al., 2003*). Despite the obvious differences, some behaviors are similar: both larvae and adults undergo forward locomotion in search of food, backward locomotion in response to noxious stimuli, and pausing in between antagonistic behaviors (*Carreira-Rosario et al., 2018*). We and others identified a neuron that, when activated, can trigger

backward locomotion in both larvae and adults (*Bidaye et al., 2014*; *Carreira-Rosario et al., 2018*; *Sen et al., 2017*), despite the obvious differences in limbless and six-legged locomotion. This neuron, named mooncrawler/moonwalker descending neuron (MDN), is present as a bilateral neuronal pair in each brain lobe, with all four MDNs having similar synaptic partners, and all four MDNs capable of eliciting backward larval locomotion in larvae (*Carreira-Rosario et al., 2018*). Larval MDNs function within a neural circuit that induces backward locomotion and coordinately arrests forward locomotion. Halting forward locomotion is achieved via activation of the Pair1 descending interneuron, which inhibits the A27h premotor neuron. Given that the A27h interneuron is required for forward locomotion, its inhibition via MDN-induced Pair1 activation prevents forward locomotion (*Carreira-Rosario et al., 2018*). Activating backward locomotion is likely to be due, in part, to MDN activation of the A18b premotor neuron, which is specifically active during backward locomotion (*Carreira-Rosario et al., 2018*). Thus, MDN-Pair1 are synaptically coupled members of a locomotor circuit in the *Drosophila* larva.

Here, we follow our previous work showing that MDN is remodeled during metamorphosis and persists into the adult (*Carreira-Rosario et al., 2018*) by asking: Is the MDN partner neuron Pair1 also maintained in the adult? Does the adult Pair1 induce an inhibition in forward locomotion, similar to its role in larvae? And, are the adult Pair1 and MDN synaptically coupled? We find that all of these questions are answered in the affirmative, showing that the core MDN-Pair1 interneuron circuit (a pair of synaptically connected interneurons) is re-established during metamorphosis despite profound neurite remodeling, and that this circuit coordinates forward/backward locomotion in both larvae and adults.

## Results

### The Pair1 neuron persists from larval to adult stages

To determine if Pair1 neurons were present in the adult, we mapped expression of a Pair1-Gal4 line (*R75C02-Gal4*) from early larval to adult stages. We identified the larval Pair1 neurons based on their characteristic cell body position in the medial subesophageal zone (SEZ), dense local ipsilateral dendritic arborizations (defined as dendritic based on enrichment for post-synapses in the TEM reconstruction of the larval Pair1 neuron; *Figure 1—figure supplement 1*), and contralateral axons descending into the ventral nerve cord (VNC) in an extremely lateral axon tract (*Carreira-Rosario et al., 2018*). Using the Pair1-Gal4 line, we could identify Pair1 neurons with this morphology at 28 and 96 hr after larval hatching (ALH; *Figure 1A,B*). The Pair1 neuron cell bodies and proximal neurites could still be observed at 24 hr after pupal formation (APF), but virtually all of the dendridic processes and descending axonal process are pruned (*Figure 1C*, only one neuron labeled). This is expected, given that many or all neurons undergo axon/dendrite remodeling during metamorphosis (*Kanamori et al., 2015*; *Truman and Reiss, 1976*; *Yaniv and Schuldiner, 2016*). At 48 hr APF, Pair1 neurons exhibited dendritic branching in the SEZ and a descending axon into the VNC, regaining morphological features similar to that of larval Pair1 neurons (*Figure 1D*). The axon innervated the T1 (prothoracic) neuropil and descended further down the VNC. These morphological features were maintained into the adult fly, where we could trace the Pair1 axon to primarily innervate the T1 neuropil (*Figure 1E*), with less extensive innervation of the mesothoracic (T2) and metathoracic (T3) neuropils.

Although we can use the Pair1-Gal4 line to track neurons with Pair1 morphological features from larva to adult, it remains possible that the Gal4 line switches off in Pair1 and switches on in a similar descending neuron at a stage in between those we assayed. To conclusively demonstrate that the larval Pair1 neuron survives into adulthood, we used a genetic technique to permanently label or 'immortalize' the larval Pair1 neurons and assay for their presence in the adult brain. Briefly, the method achieves spatial specificity by using Pair1-Gal4 to drive UAS-FLP which removes a stop cassette from nSyb-FRT-stop-FRT-LexA resulting in permanent LexA expression in Pair1-Gal4 neurons; it achieves temporal specificity (e.g. labeling only larval Pair1-Gal4+ neurons) by using a heat inducible KD recombinase to 'open' the lexAop-KDRTstopKDRT-HA reporter (see Materials and methods for additional details). Thus, a heat shock will permanently label all Pair1-Gal4+ neurons at the time

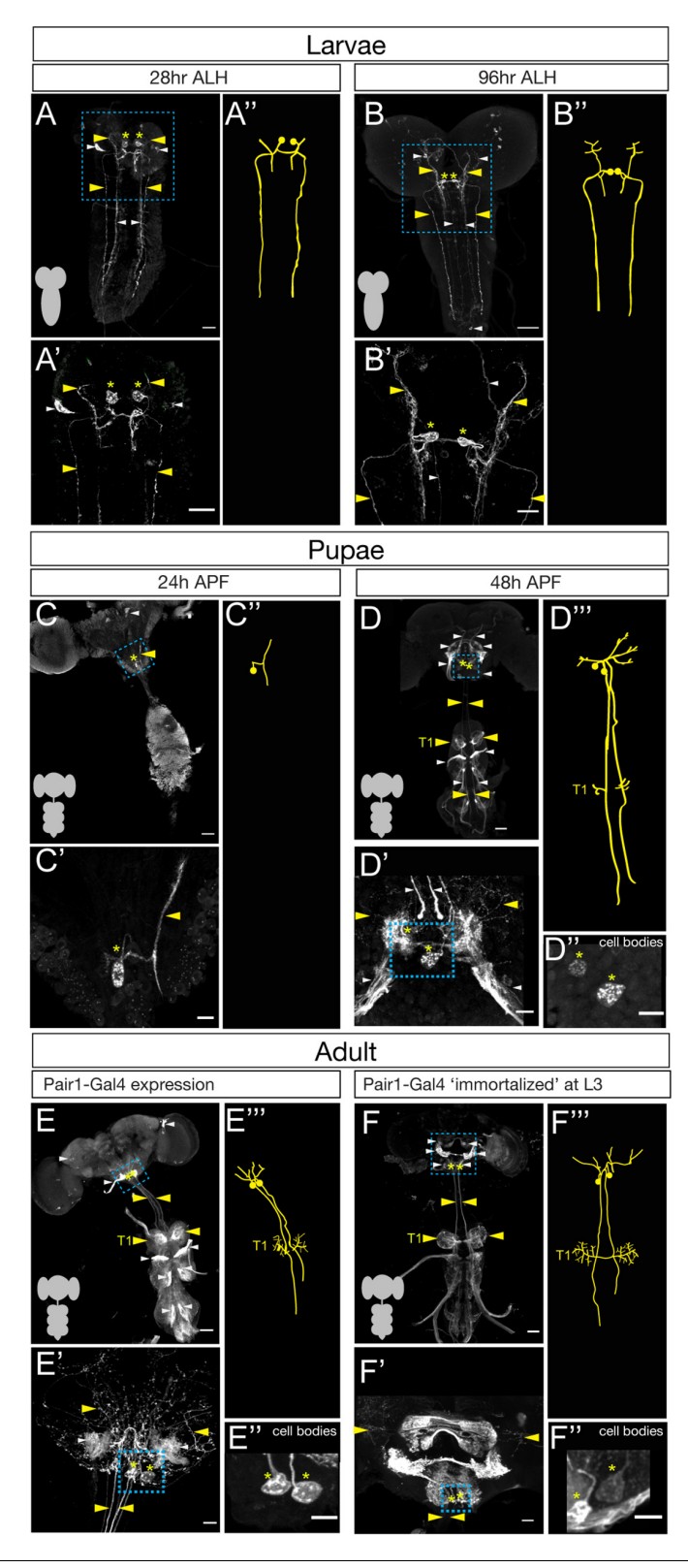

**Figure 1.** The Pair1 neuron persists from larval to adult stages. (A–B) Pair1 neurons (cell body: yellow asterisk; neurites: yellow arrowhead) in the larval CNS (gray outline) at 28 hr after larval hatching (ALH) (A) and 96 hr ALH (B). Here and in subsequent panels are maximum intensity projections of confocal sections containing the Pair1 neurons; anterior, up; dorsal view. Significant 'off-target' expression marked with white arrowheads. Scale bar, 50

*Figure 1 continued on next page*

*Figure 1 continued*

µm. (**A'–B'**) Enlargement of the brain regions boxed in A,B. Scale bar, 20 µm. (**A"–B"**) Tracing to show Pair1 neuron morphology. Genotype: *+; UAS-myr::GFP; R75C02-Gal4*. (**C–D**) Pair1 neurons (cell body: yellow asterisk; neurites: yellow arrowhead) in the pupal CNS (gray outline) at 24 hr after pupal formation (APF) (**C**) and 96 hr APF (**D**). Significant 'off-target' expression marked with white arrowheads. Scale bar, 50 µm. (**C'–D'**) Enlargement of the brain regions boxed in C, D; cell body: yellow asterisk, neurites: yellow arrowhead. Scale bar, 10 µm. (**C"**) Tracing to show Pair1 neuron morphology. (**D"**) Focal plane showing Pair1 cell bodies (region boxed in D', cell body marked with yellow asterisks). Scale bar, 10 µm. (**D"'**) Tracing to show Pair1 neuron morphology. Note that Pair1 can be followed to T1 in the 3D confocal stack but is difficult to represent here due to fasciculation of Pair1 with off-target neurons. Genotype: *+; UAS-myr::GFP; R75C02-Gal4*. (**E**) Pair1 neurons (cell body: yellow asterisk; neurites: yellow arrowhead) in the 4-day adult CNS (gray outline) Significant 'off-target' expression marked with white arrowheads. Scale bar, 50 µm. (**E'**) Enlargement of the brain region boxed in E. Scale bar, 10 µm. (**E"**) Focal plane showing Pair1 cell bodies (region boxed in E', cell body marked with yellow asterisks). Scale bar, 10 µm. (**E"'**) Tracing to show Pair1 neuron morphology. Genotype: *+; UAS-myr::GFP; R75C02-Gal4*. (**F**) Pair1 neurons (cell body: yellow asterisk; neurites: yellow arrowhead) permanently labeled at 96 hr ALH and visualized in the 4-day old adult. See Materials and methods for details. Significant 'off-target' expression marked with white arrowheads. Scale bar, 50 µm. (**F'**) Enlargement of the brain region boxed in F; Pair1 cell body: yellow asterisk; Pair1 neurites: yellow arrowhead. Scale bar, 10 µm. (**F"**) Focal plane showing Pair1 cell bodies (region boxed in F', cell body marked with yellow asterisks). Scale bar, 10 µm. (**F"'**) Tracing to show Pair1 neuron morphology. Genotype: *Hs-KD,3xUAS-FLP; 13xLexAop(KDRT.Stop)myr:smGdP-Flag/+; 13xLexAop(KDRT.Stop)myr:smGdP-V5, 13xLexAop(KDRT.Stop)myr: smGdP-HA, nSyb(FRT.Stop)LexA::p65*.

The online version of this article includes the following figure supplement(s) for figure 1:

**Figure supplement 1.** Moonwalker descending neuron (MDN) axon and Pair1 dendrite target the same neuropil in the larval brain.

of heat shock. We immortalized Pair1 neurons in the larva, and assayed expression in the adult, and observed the two bilateral Pair1 neurons, based on characteristic medial SEZ cell body position, local ipsilateral arbors, and contralateral descending axons that preferentially innervate the prothoracic neuropil (*Figure 1F*). Pair1 innervation is clearer in neurons immortalized during larval stages, which reduces the off-target neuron expression in the adult VNC, and reveals an greatly enriched level of innervation in the T1 neuropil (*Figure 1F*).

The Pair1-Gal4 line is expressed in several off-target neurons in addition to Pair1. One of these, a sensory neuron from the proboscis can be removed from the adult Pair1 pattern by cutting off the proboscis a day prior to analysis (see Materials and methods) but is present at the 48 hr APF time-point (*Figure 1D,E*). In addition, there are off-target neurons that innervate all three thoracic neuro-pils (T1-T3), obscuring Pair1 innervation (*Figure 1E*). We took advantage of the sparse labeling of the immortalization genetics and found brains that maintained preferential targeting of Pair1 to the prothoracic neuropil but lacked T1-T3 off-target innervation, confirming that they are indeed off-tar-get neurons (*Figure 1F–F'''*).

## Pair1 neurons maintain the same molecular profile from larval to adult stages

If Pair1 neurons persist from larva to adult, they may express the same transcription factor (TF) pro-file at both stages. We screened a small collection of TF markers for expression in the larval and adult Pair1 neurons, and in all cases we found identical expression (*Figure 2A–N*). Larval and adult Pair1 neurons expressed Hunchback (Hb), Sex combs reduced (Scr), and Bicoid (Bcd); but did not express Visual system homeobox (Vsx1) or Nab (*Figure 2A–N*). However, there were many other Scr/Bcd/Hb triple-positive neurons in the SEZ (*Figure 2O,P*), showing that additional factors would be necessary to uniquely specify Pair1 identity. These results support the conclusion that Pair1 per-sists from larva to adult, maintaining both molecular and morphological features, and raises the interesting possibility that the three TFs (Hb, Bcd, Scr) may be part of a molecular code that directs both larval and adult Pair1 morphology and/or connectivity.

## Pair1 activation arrests forward locomotion in adults

We previously showed that larval MDN persists in adults and can induce backward locomotion at both stages despite the obvious difference in motor output – limbless crawling versus six-legged

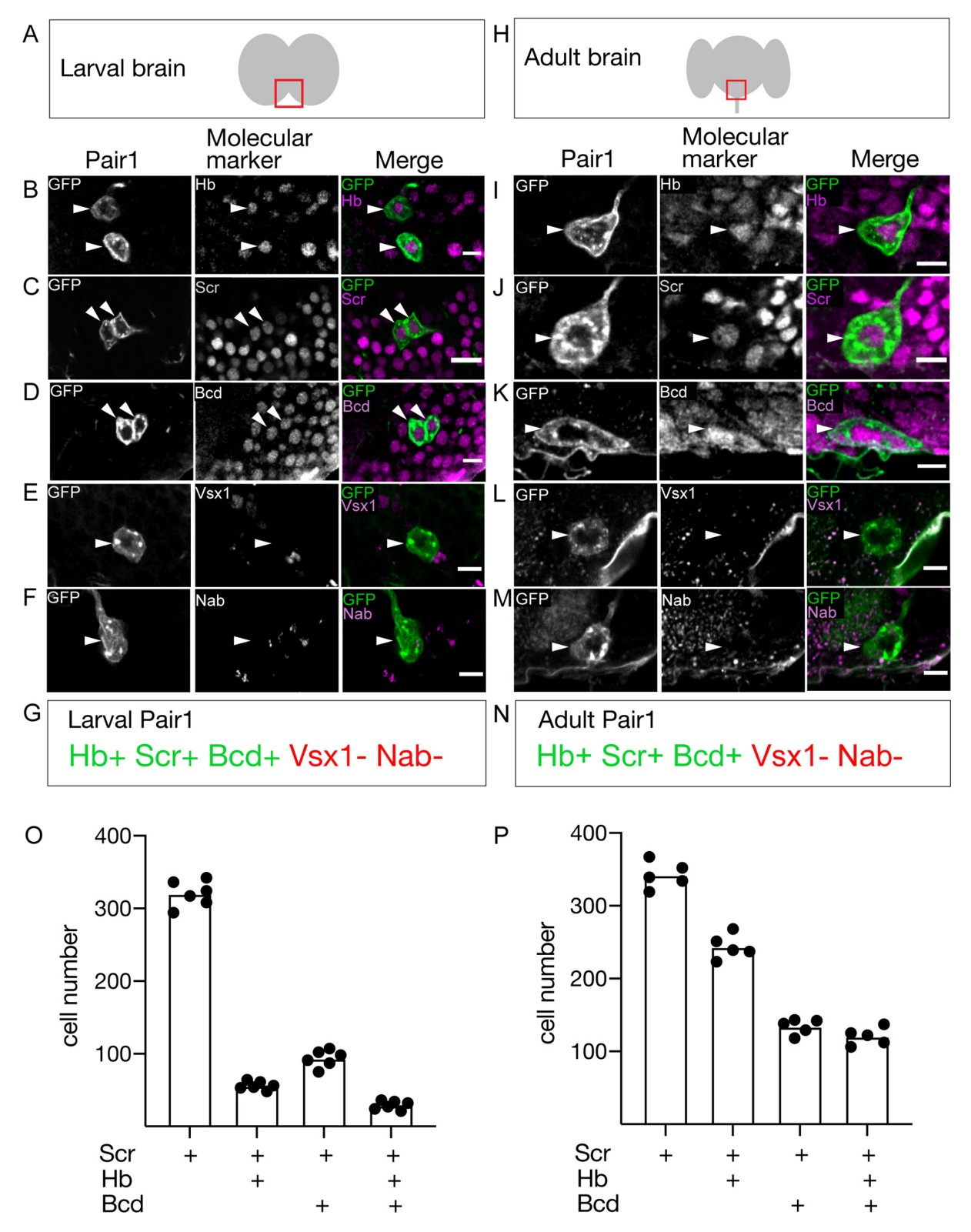

**Figure 2.** The Pair1 neuron expresses the same molecular markers at larval and adult stages. (**A**) Schematic of the larval brain showing region of Pair1 neurons (red box) enlarged in panels below. Anterior up, dorsal view. (**B–G**) Larval Pair1 neurons (left column), indicated markers (middle column), and merge (right column) at 28 hr after larval hatching (ALH). In some cases the second Pair1 neuron is out of the focal plane, but both Pair1 neurons always have the same gene expression profile. Markers detect the following transcription factors: Hb, Hunchback; Scr, Sex combs reduced; Bcd, Bicoid; Vsx1,

*Figure 2 continued on next page*

*Figure 2 continued*

Visual system homeobox 1; and Nab. Scale bar, 5 μm. (**G**) Summary: marker expression matches that in adults. Genotype: *+; UAS-myr::GFP; R75C02-Gal4*. (**H**) Schematic of the adult brain showing region of Pair1 neurons (red box) enlarged in panels below. Anterior up, dorsal view. (**I–N**) Adult Pair1 neurons (left column), indicated markers (middle column), and merge (right column) in 4-day old adult. Scale bar, 5 μm. (**N**) Summary: marker expression matches that in larvae. Genotype: *+; UAS-myr::GFP; R75C02-Gal4*. (**O–P**) The number of cells expressing Scr (first column), Scr/Hb (second column), Scr/Bcd (third column), and Scr/Hb/Bcd (fourth column) in larvae (**O**) and adults (**P**). n = 5–6 whole brains.

The online version of this article includes the following source data for figure 2:

**Source data 1.** Raw cell counts - *Figure 2*.

walking (*Carreira-Rosario et al., 2018*). This raised the question of whether the adult Pair1 neuron also maintains its larval function, that is, to pause forward locomotion. To test this hypothesis, we used Pair1-Gal4 to express the red light-gated cation channel CsChrimson (Chrimson) to activate Pair1 neurons in the adult. Experimental flies were fed all-*trans* retinal (ATR; required for Chrimson function) whereas control flies were fed vehicle only.

Control flies exposed to red light did not pause or arrest forward locomotion, did not show an increased probability of pausing, and did not have a decrease in distance traveled during the stimulus interval. In contrast, experimental flies expressing Chrimson in Pair1 neurons showed a near-complete arrest of forward locomotion, an increased probability of pausing, and a reduced distance traveled during the stimulus interval (*Figure 3A–C*; *Figure 3—figure supplement 1*). These effects were reversed after turning off the red light, with the exception of a slightly reduced distance traveled, likely due to a lingering physiological effect of the 30 s Pair1 activation (*Figure 3A,C*). Pair1 activation resulted in an increase in immobile flies (*Figure 3E*) and a corresponding decrease in whole body translocation (*Figure 3F*, defined as 'large movements'). Importantly, Pair1 activation did not prevent small body part movements such as those involved in grooming (*Figure 3G*, defined as 'small movements'). Note that Pair1-Gal4 off-target expression is common but variable from fly to fly, whereas its expression in Pair1 neurons is fully penetrant; because the Chrimson-induced behavior is also fully penetrant, we conclude that the arrest in forward locomotion is due to Chrimson activation of the Pair1 neurons. We conclude that Pair1 activation prevents a single behavior – forward locomotion – but does not produce general paralysis or interfere with non-translocating limb movements.

## MDN and Pair1 are synaptic partners during adulthood

Given that MDN and Pair1 are synaptic partners in the larvae (*Figure 1—figure supplement 1*), MDN and Pair1 persist into adulthood (*Figures 1* and *2*), and MDN and Pair1 both regulate the same behavior in larvae and adults (*Figure 3*; *Carreira-Rosario et al., 2018*), we hypothesized that MDN and Pair1 may also be synaptic partners during adulthood. To test this hypothesis, we used the MDN-LexA and Pair1-Gal4 to label MDN and Pair1 neurons individually in the same animal (*Figure 4A,B*). We observed MDN and Pair1 neurites in close proximity to each other (*Figure 4C–E*).

To determine if MDN and Pair1 are synaptic partners in this region of neuropil, we utilized t-GRASP (targeted GFP reconstitution across synaptic partners), an activity-independent method to label synaptic contact sites (*Shearin et al., 2018*). MDN-Pair1 connectivity was absent at 24 hr APF (*Figure 4F,G*) when neurite pruning was maximal (*Figure 1C*), but was re-established at 48 hr APF (*Figure 4H,I*) when neurite regrowth was occurring (*Figure 1D*) and maintained into the adult (*Figure 4J,K*).

Across each timepoint, control flies only expressing pre-t-GRASP in MDN did not have detectable t-GRASP signal (*Figure 4F,H and J*). Thus, MDN-Pair1 larval connectivity is established twice during development: initially established in the late embryo, and then re-established in the pupae.

## Discussion

Together with our earlier work (*Carreira-Rosario et al., 2018*), our results here show that a core interneuron circuit is preserved from larval stages into the adult. This neuronal circuit contains MDN and its monosynaptically coupled Pair1 neuron, allowing the fly to switch between antagonistic behaviors: forward versus backward locomotion. Our work raises several interesting questions: Do many other larval neural circuits persist and have similar function in adults? Do these results

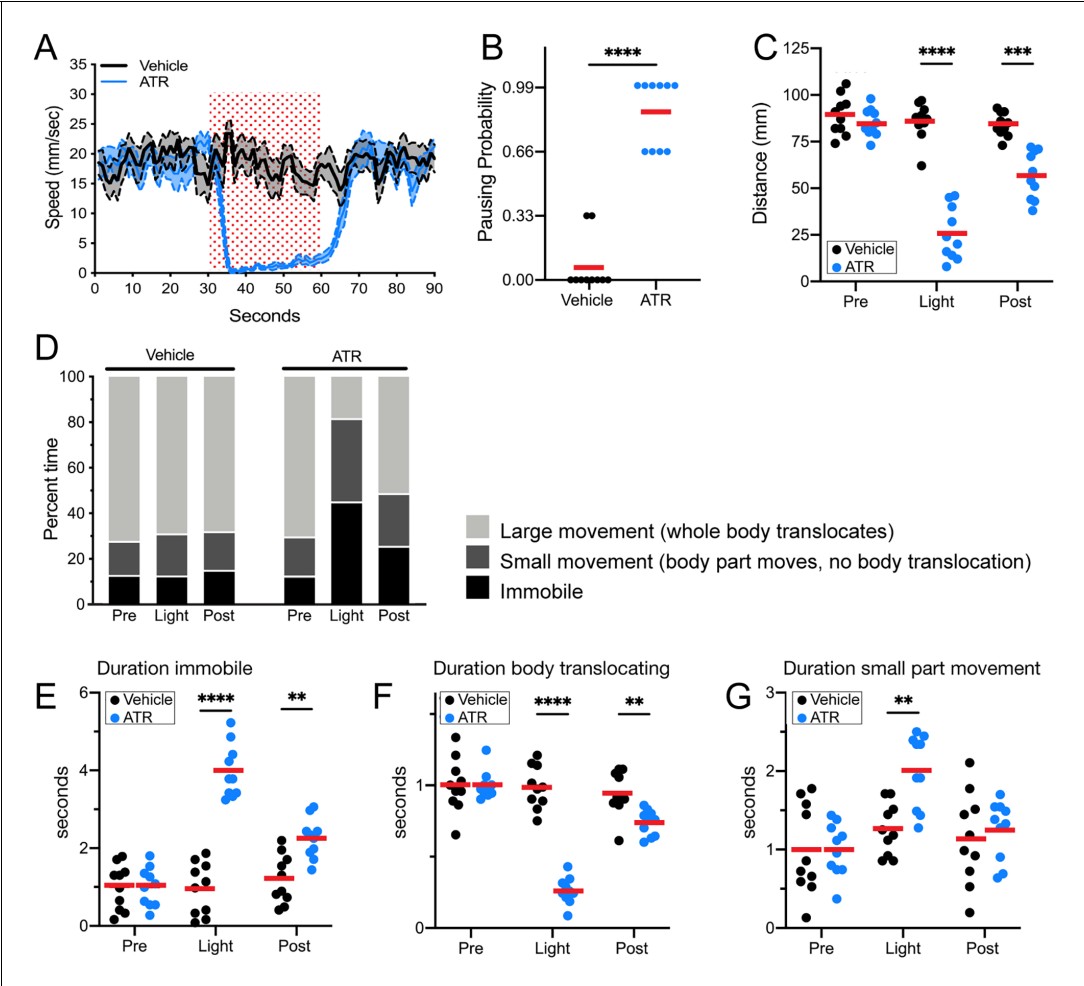

**Figure 3.** Pair1 activation for 30 s arrests forward locomotion but does not cause paralysis in adults. (A) Speed (mm/s) of adult flies expressing Chrimson in Pair1 neurons following neuronal activation (+ATR [all-*trans* retinal] , blue) or no activation (vehicle control, black) in a closed loop arena. Speed was recorded for the 30 s prior to activation, the 30 s light-induced activation (red stipple), and 30 s after activation. Mean ± SEM, n = 10. Genotype for this and all subsequent panels: *UAS-CsChrimson::mVenus; +; R75C02-Gal4*. (B) Probability of forward locomotion pausing upon light-induced Pair1 activation (ATR treatment, blue) compared to vehicle control (black). Statistics: t-test, p < 0.001; n = 10. (C) Total distance traveled pre-light stimulus ('pre'), during the light stimulus ('light') and post-light stimulus ('post') (terminology used here and in subsequent panels) of flies fed ATR (Pair1 activation, blue) compared to controls (fed vehicle, no Pair1 activation, black). Statistics: two-way ANOVA: drug treatment, $F_{(1, 18)}$ = 111.3, p < 0.0001; time, $F_{(1.867, 33.61)}$ = 47.03, p < 0.0001; interaction $F_{(2, 26)}$ = 38.24, p < 0.001; Bonferroni's multiple comparisons between drug treatments within each timepoint: pre, p > 0.9999; light, p < 0.0001; post, p = 0.0001; n = 10. (D) Percent time doing large movements (whole body translocation, light gray), small movements (body part movement but no translocation, dark gray) or no movements (immobile, black) of flies fed vehicle (left side) or ATR (right side) during each time phase (pre, light, post). (E) Normalized duration of time spent <u>immobile</u> during each timepoint (pre, light, post) for flies fed ATR (Pair1 activation, blue) compared to controls fed vehicle (black). Statistics: two-way ANOVA: drug treatment, $F_{(1, 18)}$ = 112.8, p < 0.0001; time, $F_{(1.930, 34.74)}$ = 25.55, p < 0.0001; interaction, $F_{(2, 36)}$ = 27.81, p < 0.0001; Bonferroni's multiple comparisons between drug treatments within each timepoint: pre, p > 0.9999; light, p < 0.0001; post, p = 0.0022; n = 10. (F) Normalized duration of time spent doing <u>small movements</u> during each timepoint (pre, light, post) for flies fed ATR (Pair1 activation, blue) compared to controls fed vehicle (black). Statistics: two-way ANOVA: drug treatment, $F_{(1, 18)}$ = 5.111, p = 0.036; time, $F_{(1.923, 34.62)}$ = 10.82, p = 0.0003; interaction, $F_{(2, 36)}$ = 4.225, p = 0.0225; Bonferroni's multiple comparisons between drug treatments within each timepoint: pre, p > 0.9999; light, p = 0.0022; post, p > 0.9999; n = 10. (G) Normalized duration of time spent doing <u>large movements</u> during each time phase (pre, light, post) for flies fed ATR (Pair1 activation, blue) compared to controls fed vehicle (black). Statistics: two-way ANOVA: drug treatment, $F_{(1, 18)}$ = 53.56, p < 0.0001; time, $F_{(1.869, 33.64)}$ = 53.44, p < 0.0001; interaction, $F_{(2, 36)}$ = 52.20, p < 0.0001; Bonferroni's multiple comparisons between drug treatments within each timepoint: pre, p > 0.9999; light, p < 0.0001; post, p = 0.0074; n = 10.

The online version of this article includes the following source data and figure supplement(s) for figure 3:

**Source data 1.** Raw behavior data - *Figure 3*.

**Figure supplement 1.** Pair1 activation for 4 s arrests forward locomotion but does not cause paralysis in adults in an open field arena.

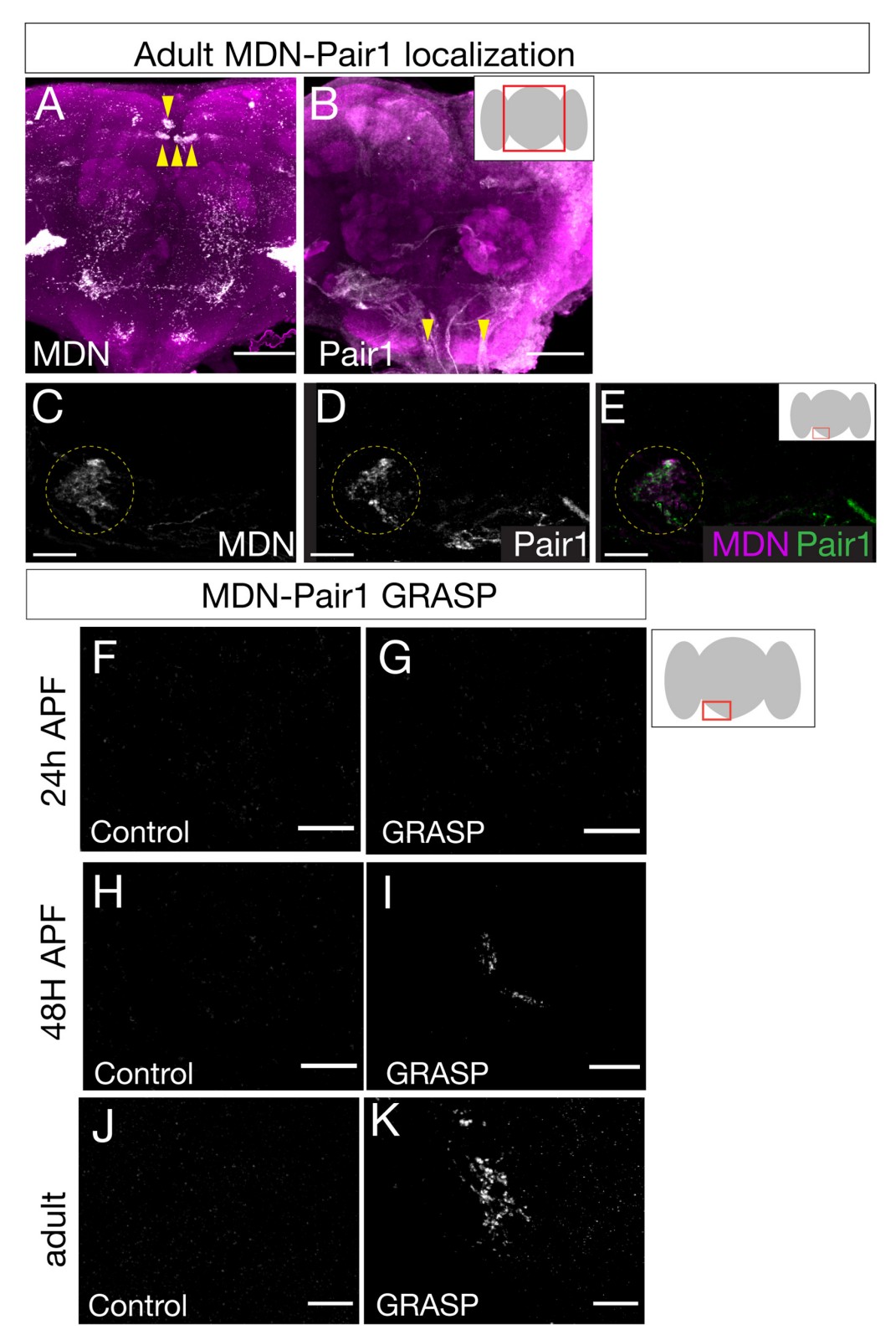

**Figure 4.** Moonwalker descending neurons (MDN) and Pair1 are synaptic partners in adults. (A–E) MDN and Pair1 show close membrane apposition. (A, B) MDN neurons (A) and Pair1 neurons (B) in the adult central brain. Neurons are in white, nc82 counterstain in magenta for whole brain orientation; cell bodies marked by yellow arrowheads. Here and in subsequent panels shows maximum intensity projection of volume; anterior, up; dorsal view. Scale bar, 50 μm. Genotype: *UAS-mCD8::RFP, LexAop-mCD8::GFP; VT044845-LexA; R75C02-Gal4.* (C–E) MDN neurites (C), Pair1 neurites

*Figure 4 continued on next page*

*Figure 4 continued*

(D), and merge (E) in the left subesophageal ganglion (red box in schematic). Scale bar, 10 μm. Genotype: *UAS-mCD8::RFP, LexAop-mCD8::GFP; VT044845-LexA; R75C02-Gal4.* (F–K) t-GRASP (targeted GFP reconstitution across synaptic partners) between MDN and Pair1. In all panels: Scale bar, 10 μm. Genotype:;; *LexAop-pre-t-GRASP, UAS-post-t-GRASP/R75C02-Gal4.* (F–G) Pupal t-GRASP at 24 hr after pupal formation (APF). (F) No detectable t-GRASP signal was observed in the subesophageal ganglion without expression of the pre-t-GRASP fragment in MDN. (G) t-GRASP signals between MDN and Pair1 were lacking in the subesophageal ganglion. (H–I) Pupal t-GRASP at 48 hr APF. (H) No detectable t-GRASP signal was observed in the subesophageal ganglion without expression of the pre-t-GRASP fragment in MDN. (I) t-GRASP signals between MDN and Pair1 were observed in the subesophageal ganglion. (J–K) Adult t-GRASP. (J) No detectable t-GRASP signal was observed in the subesophageal ganglion without expression of the pre-t-GRASP fragment in MDN. (K) t-GRASP signals between MDN and Pair1 were observed in the subesophageal ganglion.

represent a rare occurrence or a common one? Are the cues that establish MDN-Pair1 connectivity in the larvae also used to re-establish MDN-Pair1 connectivity in the adult?

How much of the larval MDN-Pair1 circuit is maintained into the adult? The larval circuit contains the MDN partners Pair1, ThDN, and A18b, and the Pair1 partner A27h (*Figure 5*; *Carreira-Rosario et al., 2018*). In addition to MDN, we show here that Pair1 is maintained. There are no Gal4 lines or markers for the ThDN neuron. Although two independent A18b Gal4 lines have extensive off-target expression, permanently labeling the Gal4 expression during larval stages consistently labels an adult neuron in the abdominal ganglion (data not shown); since MDN axons do not extend to the abdominal ganglion, it is unlikely that MDN and A18b are synaptic partners during adulthood. Additionally, the A27h interneuron, which regulates forward crawling in the larvae, undergoes apoptosis during pupal stages (data not shown) and thus cannot regulate forward walking in the adult. This is not surprising as the A27h neurons are located in the abdominal segments, which do not have a role in adult walking.

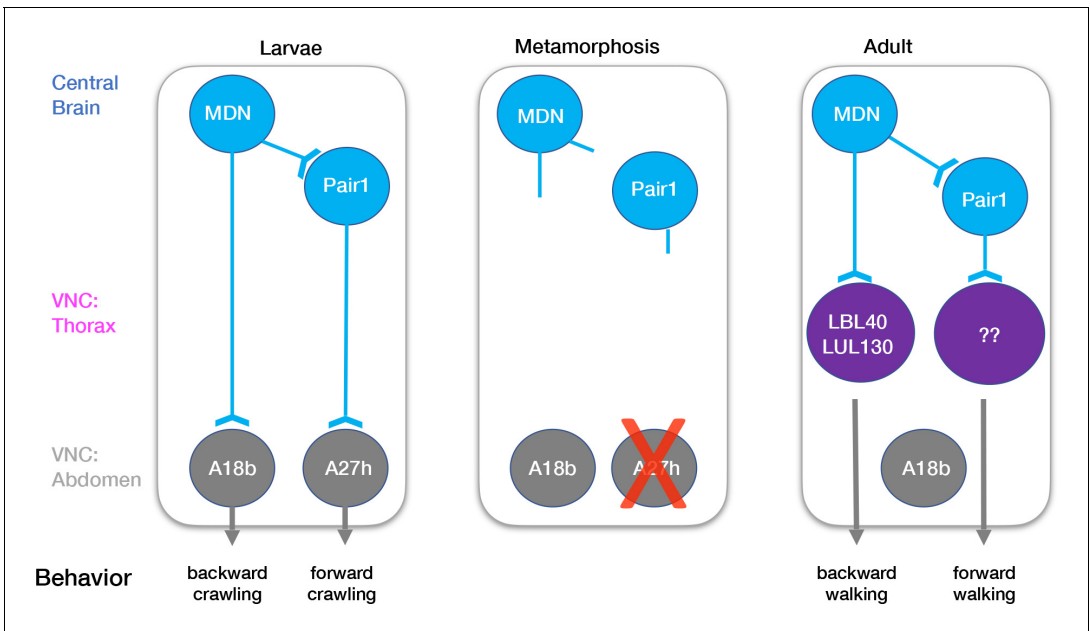

**Figure 5.** Model describing the moonwalker descending neuron (MDN)-Pair1 circuit in larval and adult stages. In the larvae, MDN and Pair1 neurons are located in the central brain. MDN and Pair1 neurons are synaptic partners. MDN neurons also extend axons into the abdominal region of the ventral nerve cord (VNC) and synapse onto A18b. A18b subsequently regulates backward crawling by synapsing onto motor neurons. Pair1 synapses onto and inhibits the pre-motor neuron A27h, which generates forward locomotion when activated. During metamorphosis, MDN and Pair1 neurons remain in the central brain, drastically prune their neurites, and survive. The A18b neuron and processes remain in the abdomen of the VNC and survives in the adult. The A27h neuron undergoes apoptosis. In the adult, synapses between MDN and Pair1 are re-established. MDN neurons also extend axons into the thoracic region of the VNC and synapse onto LBL40 and LUL130. Activation of LBL40 and LUL130 generates backward walking. MDN does not extend axons into the abdominal region of the VNC and is no longer synaptic partners with A18b. Pair1 neurons extend axons into the thoracic region of the VNC and synapse onto unknown neurons. Activation of Pair1 generates a pausing behavior, likely through the inhibition of neurons generating forward locomotion. In both larvae and adult, MDN and Pair1 neurons (blue) persist and function as a core circuit to regulate locomotion.

How much of the adult MDN-Pair1 circuit is present in the larvae? Recent work mapping the adult MDN circuit has identified over 30 VNC neurons downstream of MDN, including the LBL40 and LUL130 neurons required for hindleg backward stepping (*Figure 5*; *Feng et al., 2020*). Recent work has also identified adult neurons important for forward walking (*Bidaye et al., 2020*), but their relationship to adult MDN is unknown. In the future, it will be interesting to see if any of these adult neurons are present in the larvae, particularly those regulating forward and backward walking, and determine if they are also MDN or Pair1 target neurons.

Elegant recent work has shown that initiation of forward walking requires the forelegs, innervated by motor neurons in the prothoracic segment, whereas initiation of backward walking requires the hindlegs, innervated by motor neuron in the metathoracic segment (*Feng et al., 2020*). Similarly, adult MDN synaptic partners primarily innervate the metathoracic (T3) neuromere (*Feng et al., 2020*), a good location for inducing hindleg stepping and initiation of backward walking. A similar spatial segregation is likely to occur in the larva, where forward crawling is induced by A27h in posterior segments, and backward crawling is induced in anterior segments (*Fushiki et al., 2016*; *Tastekin et al., 2018*).

Is maintenance of neuronal circuits from larval to adult stages rare or common? Individual neurons that have similar functions in larva and adults have been identified, including select motor neurons, sensory neurons (*Consoulas et al., 2002*; *Consoulas et al., 2000*; *Levine, 1984*; *Truman, 1992*; *Weeks, 2003*), and Kenyon cells of the mushroom body (*Eichler et al., 2017*; *Li et al., 2020*). However, it remains unknown whether any of their individual synaptic partners also persist and retain the same pattern of connectivity. To date, the MDN-Pair1 circuit is the first pair of interneurons whose connectivity is established in the larva and then re-established after remodeling during metamorphosis, all while maintaining a similar function. It is currently unclear how common it is to maintain interneuron connectivity from larva to adult. If maintenance is rare, it raises the question of what is special about the MDN-Pair1 circuit? If a common occurrence, the MDN-Pair1 circuit provides an excellent model system to study a widely used mechanism.

Our work is the first, to our knowledge, to show that a pair of synaptically connected interneurons can persist from larva to adult and perform similar functions at both stages. Remarkably, both MDN and Pair1 undergo dramatic pruning and regeneration events during metamorphosis, only to reform synapses with each other following neuronal remodeling. This suggests that synapse specificity cues are maintained from the late embryo, where MDN-Pair1 connectivity is first established, into pupal stages, where MDN-Pair1 connectivity is re-established. The importance of the MDN-Pair1 interneuron circuit is highlighted by its persistence from embryo to adult, despite adapting to different sensory input and motor output at each stage. Perhaps other descending or ascending interneurons will also persist into adults, switching inputs and outputs as needed. Indeed, the idea that an interneuron circuit that is stable across developmental stages is supported by recent elegant TEM reconstruction of neural circuits at different stages of *Caenorhabditis elegans* development (*Witvliet et al., 2020*). Here, the authors conclude that 'Across maturation, the decision-making (interneuron) circuitry is maintained whereas sensory and motor pathways are substantially remodeled.' These results, together with ours, raise the possibility that preservation of interneuron circuit motifs may be functional modules that can be used adaptively with different sensorimotor inputs and outputs. The presence of this circuit motif in both flies and worms suggests that it may be an ancient evolutionary mechanism for assembling sensorimotor circuits.

## Materials and methods

**Key resources table**

| Reagent type (species) or resource | Designation | Source or reference | Identifiers | Additional information |
|---|---|---|---|---|
| Genetic reagent (*Drosophila melanogaster*) | R75C02-Gal4 | BDSC | RRID:BDSC_39886 | Short genotype: Pair1-Gal4 |

*Continued on next page*

*Continued*

| Reagent type (species) or resource | Designation | Source or reference | Identifiers | Additional information |
|---|---|---|---|---|
| Genetic reagent (*Drosophila melanogaster*) | VT044845-lexA | Gift from B Dickson, JRC | | Short genotype: MDN-LexA |
| Genetic reagent (*Drosophila melanogaster*) | UAS-myr::GFP | BDSC | RRID:BDSC_32198 | Gal4 reporter |
| Genetic reagent (*Drosophila melanogaster*) | UAS-mChrimson::mVenus | Gift from Vivek Jayaraman, JRC | | Was used to excite/depolarize neurons of interest |
| Genetic reagent (*Drosophila melanogaster*) | UAS-mCD8::RFP, LexAop-mCD8::GFP | BDSC | RRID:BDSC_32229 | Gal4 and LexA reporters |
| Genetic reagent (*Drosophila melanogaster*) | LexAop-pre-t-GRASP, UAS-post-t-GRASP | BDSC (*Shearin et al., 2018*) | RRID:BDSC_79039 | t-GRASP |
| Genetic reagent (*Drosophila melanogaster*) | Hs-KD,3xUAS-FLP; 13xLexAop(KDRT.Stop) myr:smGdP-Flag/ CyO-YFP; 13xLexAop (KDRT.Stop)myr:smGdP-V5, 13xLexAop(KDRT.Stop) myr:smGdP-HA, nSyb-(FRT.Stop)-LexA::p65/R75C02-Gal4 | This work | | Used to permanently label Gal4 pattern |
| Antibody, polyclonal | Rabbit polyclonal anti-GFP A-11122 | Thermo Fisher Scientific, Waltham, MA | RRID:AB_221569 | (1:500) |
| Antibody, polyclonal | Chicken polyclonal anti-GFP | Abcam, Eugene, OR | RRID:BDSC_13970 | (1:1500) |
| Antibody, monoclonal | Rabbit polyclonal anti-GFP (G10362) | Thermo Fisher Scientific, Waltham, MA | RRID:AB_2536526 | (1:300); used for t-GRASP |
| Antibody, monoclonal | Rat monoclonal anti-HA (3F10) | Sigma, St. Louis, MO | SKU: 11867423001 | (1:100) |
| Antibody, monoclonal | Mouse monoclonal anti-Scr | DSHB (Iowa City, IA) | RRID:AB_528462 | (1:10) |
| Antibody | Rat polyclonal anti-Bcd | Gift from John Reinitz, University of Chicago, IL | | (1:100) |
| Antibody | Guinea pig polyclonal anti-Vsx1 | Gift from Claude Desplan, NYU, New York, NY | | (1:500) |
| Antibody | Guinea pig polyclonal anti-Nab | Gift from Stefan Thor, University of Queensland, Brisbane, Australia | | (1:500) |
| Antibody | Secondary antibodies | Jackson ImmunoResearch, West Grove, PA | | (1:400); all Donkey |

## Fly husbandry

All flies were reared in a 25°C room at 50% relative humidity with a 12 hr light/dark cycle. All comparisons between groups were based on studies with flies grown, handled, and tested together.

## Fly stocks

1. *R75C02-Gal4* (Pair1 line; BDSC #39886).
2. *UAS-myr::GFP* (BDSC #32198).
3. *UAS-CsChrimson::mVenus* (Vivek Jayaraman, Janelia Research Campus).
4. *VT044845-lexA* (adult MDN line; a gift from B Dickson, Janelia Research Campus).

5. *UAS-mCD8::RFP, LexAop-mCD8::GFP;;* (BDSC #32229).
6. *LexAop-pre-t-GRASP, UAS-post-t-GRASP* (BDSC #79039).
7. Hs-KD,3xUAS-FLP; 13xLexAop(KDRT.Stop)myr:smGdP-Flag/ CyO-YFP; 13xLexAop(KDRT. Stop)myr:smGdP-V5, 13xLexAop(KDRT.Stop)myr:smGdP-HA, nSyb-(FRT.Stop)-LexA::p65/ R75C02-Gal4 (line to permanently label Pair1; Doe Lab; modified from *Ren et al., 2016*).

## Gal4 driver 'immortalization'

Immortalization flies (see genotype #7, above) were allowed to lay eggs for 4 hr. Newly hatched larvae were placed in a food vial, and at 96 hr ALH the food vial was partially submerged in a 37°C water bath for 5 min, allowing the hs-KD to act as a recombinase to remove the KDRT Stop cassette, resulting in nSyb-LexA driving HA expression permanently in the neurons expressing Pair1-Gal4 at the time of heat shock (96 hr ALH). After the heat shock, larvae in the food vial recovered at 18°C for 5 min, and then grown to adulthood at 25°C.

## Immunostaining and imaging

Standard confocal microscopy and immunocytochemistry methods were performed as previously described (*Carreira-Rosario et al., 2018*). Primary antibodies used recognize: GFP (rabbit, 1:500, Thermo Fisher Scientific, Waltham, MA; chicken, 1:1500, Abcam12970, Eugene, OR), HA (rat, 1:100, Sigma, St. Louis, MO), Hb (mouse, 1:400, AbcamF18-1G10.2), Scr (mouse, 1:10, Developmental Studies Hybridoma Bank, Iowa City, IA), Bicoid (rat, 1:100, John Reinitz, University of Chicago, IL), Vsx1 (guinea pig, 1:500, Claude Desplan, NYU, New York, NY), Nab (guinea pig, 1:500, Stefan Thor, University of Queensland, Brisbane, Australia), and t-GRASP signal (rabbit GFP G10362, 1:300, Invitrogen). Secondary antibodies were from Jackson ImmunoResearch (Donkey, 1:400, West Grove, PA). Confocal image stacks were acquired on a Zeiss 800 microscope. All images were processed in Fiji (https://imagej.new/Fiji) and Adobe Illustrator (Adobe, San Jose, CA). Images were processed as described previously (*Carreira-Rosario et al., 2018*). The primary neurites of Pair1 were traced using the Simple Neurite Tracer in Fiji.

## Cell counts

Cell counting was done manually using the 'Cell Counter' plugin in Fiji (https://imagej.new/Fiji). Only cells expressing Scr were counted.

## Adult behavioral experiment

Adult behavior was assayed using two arenas, a closed loop arena (*Figure 3*) and an open field arena (*Figure 3—figure supplement 1*). For the closed loop arena, adult female flies 1 day after eclosion were transferred to standard cornmeal fly food supplemented with 100 mL 0.5 mM ATR or 100% ethanol for 4 days (changed every 2 days). Animals, with intact wings, were starved for 4 hr and then placed in arenas and their behavior was recorded as described previously (*Carreira-Rosario et al., 2018*). Flies were exposed to low transmitted light, red light, and low transmitted light again for 30 s each. This was done three times for each animal. To calculate different parameters, the recorded videos were tracked and analyzed using the CalTech Fly Tracker (*Fontaine et al., 2009*) and JABA (*Kabra et al., 2013*). The speed, distance, and behavior reported were specific to the first trial. The reported speeds are the average speed of each second. The pausing probability was calculated as previously described (*Carreira-Rosario et al., 2018*). 'Pre' defines the 30 s prior to red light exposure, 'light' defines the 30 s of red light exposure, and 'post' defines the 30 s after red light exposure. Immobile movements were defined as the fly not translocating and not moving other body parts. Small movements were defined as the fly not translocating but moving body parts (i.e. grooming, moving wings). Large movements were defined as the fly translocating its body. All behavior measures were normalized by dividing them by the group average 'pre' values.

For the open field arena, adult flies were fed ATR and vehicle as described above. Three animals were placed in a circular arena with a diameter of 14.5 cm and height of 0.5 cm. After 5 min for environmental acclimation, animal behavior was recorded at 25 FPS using a Basler acA2040-25gm GigE camera under infrared light for 4 s followed by 4 s under red light and another 4 s under infrared light, as described previously (*Risse et al., 2013*). The was repeated three times, and tracked and analyzed as described above.

### t-GRASP

Synapse establishment was investigated via t-GRASP. Flies with MDN-LexA and Pair1-Gal4 driving expression of t-GRASP (see genotype #6 above) were reared at 25°C and dissected at 24 hr APF, 48 hr APF, and 4 days post-eclosion. Control flies lacked MDN-LexA.

### Statistics

All statistical analysis (t-test, one-way and two-way ANOVA with Bonferroni's multiple comparison tests) were performed with Prism 9 (GraphPad Software, San Diego, CA). Numerical data in graphs show individual measurements (animals), means (represented by red bars) or means ± SEM (dashed lined), when appropriate. The number of replicates (n) is indicated for each data set in the corresponding legend.

## Acknowledgements

We thank John Reinitz, Claude Desplan, and Stefan Thor for antibodies; Barry Dickson, Matthieu Loius, and Vivek Jayaraman for fly stocks. Transgenic lines were generated by BestGene (Chino Hills, CA) or Genetivision (Houston, TX). Stocks obtained from the Bloomington *Drosophila* Stock Center (NIH P40OD018537) were used in this study. We thank Dr Sen-Lin Lai for the immortalization fly stock, and Sen-Lin Lai, Emily Heckman, and Arnaldo Carreira-Rosario for comments on the manuscript. Funding was provided by HHMI (CQD, KML).

## Additional information

### Competing interests

Chris Q Doe: Reviewing editor, *eLife*. The other author declares that no competing interests exist.

### Funding

| Funder | Grant reference number | Author |
|---|---|---|
| Howard Hughes Medical Institute | | Chris Q Doe |
| National Institutes of Health | HD27056 | Kristen Lee |
| NIH | P40OD018537 | Chris Q Doe |

The funders had no role in study design, data collection and interpretation, or the decision to submit the work for publication.

### Author contributions

Kristen Lee, Conceptualization, Resources, Data curation, Software, Formal analysis, Validation, Investigation, Visualization, Methodology, Writing - original draft, Writing - review and editing; Chris Q Doe, Conceptualization, Data curation, Supervision, Funding acquisition, Writing - original draft, Project administration, Writing - review and editing

### Author ORCIDs

Kristen Lee  https://orcid.org/0000-0003-4527-6468
Chris Q Doe  https://orcid.org/0000-0001-5980-8029

### Decision letter and Author response

Decision letter https://doi.org/10.7554/eLife.69767.sa1
Author response https://doi.org/10.7554/eLife.69767.sa2

## Additional files

### Supplementary files
• Transparent reporting form

### Data availability
All data generated or analysed during this study are included in the manuscript and supporting files.

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
