## [Decision Letter]

**Acceptance summary:**

This paper will be of interest to scientists interested in comparative neural circuits and how elements of circuit control are conserved across changes in development (or presumably evolution). The authors use compelling genetic experiments to show that Pair1 neurons, a synaptic partner of moonwalker descending neurons (MDN), are conserved between larval and adult stages, retracting during pupation and re-forming connections and much of their behavioral function in adulthood. This adds to our growing understanding of circuit conservation across insect brains.

**Decision letter after peer review:**

Thank you for submitting your article "A locomotor neural circuit persists and functions similarly in larvae and adult *Drosophila*" for consideration by *eLife*. Your article has been reviewed by 2 peer reviewers, and the evaluation has been overseen by Ronald Calabrese as the Senior and Reviewing Editor. The reviewers have opted to remain anonymous.

Essential revisions:

This is a solid piece of work but the advance as presented is limited.

1. A primary concern is that there is no broader consideration of other connections of MDN in the larval transformation to adult. The focus on one specific connection being preserved may not represent the overall scale of remodeling during metamorphosis. There are a variety of experiments proposed that could provide information on this issue and give stronger insight into what types of connections are preserved/reconnected. For example, the trans-tango experiments proposed, even if they don't allow the authors to identify every connected cell-type, could provide a sense of scale. Part of this could be fixed by text revisions if specific experiments are not feasible.

2. A second concern is the breadth of TF staining suggesting that the specific TFs observed may not be controlling connectivity.

3. A third concern is the framing of the MDN-Pair1 connection as a decision-making circuit. This concern is best addressed by text revision. The detailed comments of the reviewers will provide specific reasoning/recommendations to the authors.

*Reviewer #1:*

This study from Lee and Doe examines the conservation of morphology, function, and connectivity in a *Drosophila* neural circuit that promotes backward locomotion. The study builds on a previous *eLife* paper from this group, Carreira-Rosario et al. 2018, which showed that a set of descending neurons- the Moonwalker Descending Neurons (MDN)- were conserved between larval and adult stages and performed a similar function in initiating backward locomotion, despite the very different locomotor strategies of crawling larvae and walking adults. Here, the authors examine a second neuron type- Pair1- that is downstream of MDN and inhibits forward locomotion. They first use a genetic immortalization strategy to show that the morphology of this neuron is similar between larvae and adults, although many of its processes are pruned during pupation. They next show that a set of transcription factor markers in these neurons are conserved between larval and adult stages. Finally these show that optogenetic activation of Pair1 in adults generates a similar stopping phenotype as in larvae, and that MDN and Pair 1 and synaptically connected (via GRASP) in adults as well. The authors conclude that these neurons form a core "decision-making" circuit that is conserved between developmental stages, and speculate that the continued expression of the same transcription factors may allow these neurons to re-form synapses after pupation. Overall this is a clear study that provides new data on which parts of the fly locomotor circuit are maintained across developmental stages. Together with on-going work in other parts of the fly brain, and across insect species, I expect this work to contribute to our understanding of how neural circuits become adapted through genetic and developmental mechanisms to different organismal niches.

The data are generally strong and convincing. One concern I have is that the transcription factor stains in Figure 2 show that most of the surrounding cell bodies share the same pattern of expression for the 5 TFs examined, with Hb, Scr, and Bcd highly expressed and Vsx1 and Nab absent. This might impact the authors' model that TF expression patterns in Pair1 allow it to make correct synaptic connections with MDN. The authors should address whether they expect these surrounding neuron to make similar synaptic connections, or whether connectivity is determined by a larger group of TFs other than those shown here. Experiments manipulating TF expression and looking at effects on connectivity would strength this part of the authors' model.

An additional concern is that the MDN-Pair1 neurons are referred to multiple time in the Discussion as a "decision-making circuit." However this term is poorly defined. The authors should specify what they mean by this term and how they would distinguish a "decision-making circuit" from other central circuits. More broadly, it would be helpful to be specific about what central circuit features they would expect to be conserved between larvae and adults or to state clearly if there too little data yet to make broad predictions of this type.

*Reviewer #2:*

This paper asks whether a neural connection that exists in the brain of the *Drosophila* larva also forms in the adult brain and serves a similar function. Past work (Carreira-Rosario et al., 2018) showed that in larvae, the mooncrawler descending neuron (MDN) drives the transition from forward to backward crawling in part through its connection onto the bilateral pair of Pair1 neurons. Here the authors find that Pair1 Gal4 lines label a pair of adult descending neurons with cell bodies in the gnathal ganglia which express the same developmental markers as the larval Pair1 neurons. Positive GRASP staining using MDN and Pair1 Gal4 lines suggests the adult neurons are synaptically connected. Furthermore, optogenetic activation of Pair1 neurons causes adult flies to stop walking or moving, similar to the effect found for Pair1 activation in larvae. They show that Pair1 neurons undergo dramatic remodeling of their axonal arbors in the ventral nerve cord (VNC).

This paper extends the work of Carreira-Rosario et al., (2018), by following the Pair1 neurons through development. As a conceptual advance, it is an interesting finding that Pair 1 neurons play similar functional roles in adults as in larvae, even though the downstream targets must be entirely different. Pair1 neurons were shown to form inhibitory synapses onto premotor neurons A27h in posterior abdominal segments, which themselves drive muscle contraction and initiate forward crawling. A27h undergo apoptosis during pupal development, abdominal segments lose their role in locomotion, and Pair1 develops extensive arbors in the adult T1 where locomotion is likely initiated.

The work addresses important and interesting questions, but its scope is limited. While the authors identify some similarities in connectivity and function of Pair1 neurons across metamorphosis, the characterization of the Pair1 neurons is far from complete.

I feel that two major revisions would improve the paper. First, there are some simple experiments that would help broaden the scope and impact of the study. The authors could survey the downstream partners of Pair1 in the T1 thoracic neuromere using Trans-tango. They could also ask whether silencing Pair1 (e.g. driving Kir2.1 expression) causes behavioral deficits, to test whether pair 1 neurons are necessary for the switch between forward and backward walking.

The paper would also benefit from a deeper exploration of the degree to which the preservation of MDN to Pair1 connectivity is representative or extraordinary. For example, are there connectivity motifs from the larvae that the authors did not find in the adult? Including these negative results would help provide context. Many neurons (e.g. thoracic motor neurons) gain a role in adulthood, others lose a role (e.g. abdominal motor neurons no longer control "locomotion"). What would one expect if it were possible to compare all the inputs and outputs of Pair1, or other fly neurons?

Given that the paper builds on a previous study, it could also be improved by editing for concision. There are some sections that distract from the key take-home message. For example, the authors state that this circuit plays a role in making decisions. I understand the point, that switching from forward to backward locomotion indicates that a decision has been computed, but it begs questions that are not addressed, such as what exactly constitutes a decision? Where is the decision being made? What mechanism, like a threshold, or competition between neurons, underlie when a decision takes place? Can adult flies stop (Pair1 activation by MDN) without moving backwards (LBL40, LUL130)? If the paper focuses on the developmental questions, provides background on what other interesting motifs are seen as circuits remodel, and position the results within those broader themes, I feel I would have a clearer sense of the novelty of these findings.

Issues:

Pair1 optogenetic activation causes larvae to pause briefly, whereas adults stop. The authors speculate it's due to circuitry. Could temporal differences in Channelrhodopsin kinetics and light protocols explain this?

Some edits:

This neuron, named Moonwalker/Mooncrawler Descending Neuron

(MDN) is present in two bilateral pairs per brain lobe, – Awkward phrasing.

"Halting forward locomotion is done via activation of the Pair1 descending

interneuron, which inhibits the A27h premotor neuron, to prevent it from inducing forward locomotion (Carreira-Rosario et al., 2018)." – Unclear.

"We find that all of these questions are answered in the affirmative, showing that the core MDN-Pair1 decision-making circuit (a pair of synaptically-connected interneurons) persists from larva to adult, despite profound remodeling during metamorphosis, and that this circuit coordinates forward/backward locomotion in both larvae and adults." – see point above.

"induce backward locomotion via the coordinate arrest of forward locomotion" – unclear.

77 – "but virtually all of the dendridic processes and descending axonal process had been pruned (Figure 1C, only one neuron labeled)."

107 – "We conclude that the Pair1 neurons are present from larval to adult stages, and that the Pair1 neurons are enriched for postsynaptic partners in the T1

neuromere." – "enriched" is a strange word in this context.

[Editors' note: further revisions were suggested prior to acceptance, as described below.]

Thank you for submitting your article "A locomotor neural circuit persists and functions similarly in larvae and adult *Drosophila*" for consideration by *eLife*. Your article has been reviewed by 2 peer reviewers, and the evaluation has been overseen by a Reviewing Editor and Ronald Calabrese as the Senior Editor. The reviewers have opted to remain anonymous.

Essential revisions:

Please shorten the Discussion and remove the Trans-Tango data as suggested by Reviewer #2.

*Reviewer #1:*

The authors have addressed my concerns.

*Reviewer #2:*

The authors have revised their manuscript, A locomotor neural circuit persists and functions similarly in larvae and adult *Drosophila*, which details the persistence of a circuit motif across metamorphosis, undergoing remodeling while maintaining a similar function. In my opinion, the writing has been significantly improved by raising and supporting the notion that remodeling on this scale has not been observed, both in the introduction and the discussion. The authors also found a positive, provocative way to suggest MDN-Pair1 participate in decision-making by citing recent results in *C. elegans*. The discussion is currently long and slightly repetitive with the intro and text; I would recommend another edit to shorten. For example, a paragraph dedicated to the identification of transcription factors in the discussion is unnecessary. Finally, the paper reads more clearly as an extension of Carreira-Rosario et al., 2018, which makes it more appropriate for an *eLife* Advance.

I appreciate the addition of the Trans-Tango data, but they are of limited utility since downstream neurons are not clearly labelled or identifiable, and what labelling does exist could indicate downstream partners of the sensory neurons labeled by the Pair1 Gal4 line. I noticed in the results that the authors dissected the adult CNS 4 days post-eclosion. However, it generally takes much longer for the trans-tango signal to appear. The original paper (Talay et al., 2017) used 15-20 day old flies. I recommend that the authors' either repeat the experiments with a positive control to ensure that trans-tango is working, or remove them from the manuscript entirely.

---

## [Author Response]

Essential revisions:This is a solid piece of work but the advance as presented is limited.1. A primary concern is that there is no broader consideration of other connections of MDN in the larval transformation to adult. The focus on one specific connection being preserved may not represent the overall scale of remodeling during metamorphosis. There are a variety of experiments proposed that could provide information on this issue and give stronger insight into what types of connections are preserved/reconnected. For example, the trans-tango experiments proposed, even if they don't allow the authors to identify every connected cell-type, could provide a sense of scale. Part of this could be fixed by text revisions if specific experiments are not feasible.

We add several findings to address this interesting question. First, we document Pair1 trans-tango expression, showing Pair1 output neurons for both larva and adults. Pair1 has relatively few downstream partners at either larvae and adult stages (new Figure 1 – Supplement 2). This is substantiated by TEM analysis in larvae, where we find only two Pair1 output neurons that are bilateral with >1 synapse: the A27h and DN_mx (in addition to a lot of unknown neuron fragments). In the adult, none of the Pair1 output neurons are in similar anatomical locations as larval Pair1 output neurons. Thus, it is unlikely for there to many additional Pair1 output neurons that persist from larva to adult. Second, at least one of these Pair1 downstream partners, A27h, is not found in the adult (updated Figure 5). These data suggests that an interneuron-interneuron connection persisting from larvae to adulthood may be unusual, rather than common.

2. A second concern is the breadth of TF staining suggesting that the specific TFs observed may not be controlling connectivity.

We agree that Pair1 does not uniquely express Hb/Bcd/Scr. Quantification has been added to show that within the SEZ approximately 10-40% of cells are triple-positive neurons (updated Figure 2). Interestingly, only the SEZ has triple-positive neurons.

3. A third concern is the framing of the MDN-Pair1 connection as a decision-making circuit. This concern is best addressed by text revision. The detailed comments of the reviewers will provide specific reasoning/recommendations to the authors.

We have removed all mentions of the a “decision-making” circuit from the text.

Reviewer #1:This study from Lee and Doe examines the conservation of morphology, function, and connectivity in a *Drosophila* neural circuit that promotes backward locomotion. The study builds on a previous eLife paper from this group, Carreira-Rosario et al. 2018, which showed that a set of descending neurons- the Moonwalker Descending Neurons (MDN)- were conserved between larval and adult stages and performed a similar function in initiating backward locomotion, despite the very different locomotor strategies of crawling larvae and walking adults. Here, the authors examine a second neuron type- Pair1- that is downstream of MDN and inhibits forward locomotion. They first use a genetic immortalization strategy to show that the morphology of this neuron is similar between larvae and adults, although many of its processes are pruned during pupation. They next show that a set of transcription factor markers in these neurons are conserved between larval and adult stages. Finally these show that optogenetic activation of Pair1 in adults generates a similar stopping phenotype as in larvae, and that MDN and Pair 1 and synaptically connected (via GRASP) in adults as well. The authors conclude that these neurons form a core "decision-making" circuit that is conserved between developmental stages, and speculate that the continued expression of the same transcription factors may allow these neurons to re-form synapses after pupation.Overall this is a clear study that provides new data on which parts of the fly locomotor circuit are maintained across developmental stages. Together with on-going work in other parts of the fly brain, and across insect species, I expect this work to contribute to our understanding of how neural circuits become adapted through genetic and developmental mechanisms to different organismal niches.The data are generally strong and convincing. One concern I have is that the transcription factor stains in Figure 2 show that most of the surrounding cell bodies share the same pattern of expression for the 5 TFs examined, with Hb, Scr, and Bcd highly expressed and Vsx1 and Nab absent. This might impact the authors' model that TF expression patterns in Pair1 allow it to make correct synaptic connections with MDN. The authors should address whether they expect these surrounding neuron to make similar synaptic connections, or whether connectivity is determined by a larger group of TFs other than those shown here. Experiments manipulating TF expression and looking at effects on connectivity would strength this part of the authors' model.

See general comment 2, above. We agree that Pair1 does not uniquely express Hb/Bcd/Scr. Quantification has been added to show that within the SEZ approximately 10-40% of cells are triple-positive neurons (new Figure 2).

An additional concern is that the MDN-Pair1 neurons are referred to multiple time in the Discussion as a "decision-making circuit." However this term is poorly defined. The authors should specify what they mean by this term and how they would distinguish a "decision-making circuit" from other central circuits. More broadly, it would be helpful to be specific about what central circuit features they would expect to be conserved between larvae and adults or to state clearly if there too little data yet to make broad predictions of this type.

We agree with this concern and have updated the text to exclude the phrase “decision-making circuit”.

Reviewer #2:This paper asks whether a neural connection that exists in the brain of the *Drosophila* larva also forms in the adult brain and serves a similar function. Past work (Carreira-Rosario et al., 2018) showed that in larvae, the mooncrawler descending neuron (MDN) drives the transition from forward to backward crawling in part through its connection onto the bilateral pair of Pair1 neurons. Here the authors find that Pair1 Gal4 lines label a pair of adult descending neurons with cell bodies in the gnathal ganglia which express the same developmental markers as the larval Pair1 neurons. Positive GRASP staining using MDN and Pair1 Gal4 lines suggests the adult neurons are synaptically connected. Furthermore, optogenetic activation of Pair1 neurons causes adult flies to stop walking or moving, similar to the effect found for Pair1 activation in larvae. They show that Pair1 neurons undergo dramatic remodeling of their axonal arbors in the ventral nerve cord (VNC).This paper extends the work of Carreira-Rosario et al., (2018), by following the Pair1 neurons through development. As a conceptual advance, it is an interesting finding that Pair 1 neurons play similar functional roles in adults as in larvae, even though the downstream targets must be entirely different. Pair1 neurons were shown to form inhibitory synapses onto premotor neurons A27h in posterior abdominal segments, which themselves drive muscle contraction and initiate forward crawling. A27h undergo apoptosis during pupal development, abdominal segments lose their role in locomotion, and Pair1 develops extensive arbors in the adult T1 where locomotion is likely initiated.The work addresses important and interesting questions, but its scope is limited. While the authors identify some similarities in connectivity and function of Pair1 neurons across metamorphosis, the characterization of the Pair1 neurons is far from complete.I feel that two major revisions would improve the paper. First, there are some simple experiments that would help broaden the scope and impact of the study. The authors could survey the downstream partners of Pair1 in the T1 thoracic neuromere using Trans-tango. They could also ask whether silencing Pair1 (e.g. driving Kir2.1 expression) causes behavioral deficits, to test whether pair 1 neurons are necessary for the switch between forward and backward walking.

We thank the reviewer for suggesting these experiments. We have included data showing a small number of Pair1 downstream partners labeled by Trans-tango in the adult (Figure 1 – Supplement 2). Given that silencing Pair1 neurons in larvae did not lead to any behavioral differences (Carreira-Rosario et al., 2018), we did not test this in the adult.

The paper would also benefit from a deeper exploration of the degree to which the preservation of MDN to Pair1 connectivity is representative or extraordinary. For example, are there connectivity motifs from the larvae that the authors did not find in the adult? Including these negative results would help provide context. Many neurons (e.g. thoracic motor neurons) gain a role in adulthood, others lose a role (e.g. abdominal motor neurons no longer control "locomotion"). What would one expect if it were possible to compare all the inputs and outputs of Pair1, or other fly neurons?

We tested multiple neurons from the larval MDN circuit for a role in the adult MDN circuit, and have added that data to the manuscript (updated Figure 5). From the larval MDN circuit, the A27h neurons do not persist into adulthood, and the A18b neurons are present in the adult but are not part of the MDN circuit. Yet it remains unclear whether maintaining anatomical and behavioral similarity from larva to adult is a rare or common phenomena. Either way is interesting: if rare, it leads to the question of how does persistent connectivity survive metamorphosis in the MDN-Pair1 circuit but not other circuits; if common, it provides an excellent model system to study a widely used mechanism. This is an important question for future study!

Given that the paper builds on a previous study, it could also be improved by editing for concision. There are some sections that distract from the key take-home message. For example, the authors state that this circuit plays a role in making decisions. I understand the point, that switching from forward to backward locomotion indicates that a decision has been computed, but it begs questions that are not addressed, such as what exactly constitutes a decision? Where is the decision being made? What mechanism, like a threshold, or competition between neurons, underlie when a decision takes place? Can adult flies stop (Pair1 activation by MDN) without moving backwards (LBL40, LUL130)? If the paper focuses on the developmental questions, provides background on what other interesting motifs are seen as circuits remodel, and position the results within those broader themes, I feel I would have a clearer sense of the novelty of these findings.

Thank you, we agree with these comments. We have removed all mention of decision-making circuits, because (as the reviewer points out) we have not provided sufficient context for this claim. Regarding novelty, we emphasize that this is the first interneuronal circuit known to persist from larva to adult while maintaining a similar function. We mention a few sensory-motor reflex circuits that are maintained, and we discuss whether larval Kenyon cells maintain the same inputs (DANs) and outputs (MBONs). Although all three classes of interneurons are maintained from larva to adult, it remains unknown whether the same neuronal partners are maintained, e.g. whether a specific pair of DAN-KC neurons in the larva will reform the same pairing in the adult.

Issues:Pair1 optogenetic activation causes larvae to pause briefly, whereas adults stop. The authors speculate it's due to circuitry. Could temporal differences in Channelrhodopsin kinetics and light protocols explain this?

We thank the reviewer for this comment. We have added some discussion on the changes in the behavior between larvae and adult. Many differences do exist between the larvae and adult experiments – in addition to the differences in Channel rhodopsin kinetics and the light protocol, the adult flies also have a cuticle that makes light more difficult to pass through, whereas larvae are transparent. Even with the complicated nature of these experiments, the result that the adult flies pause for the entire duration of red-light exposure is very striking, making the differences in circuitry between larvae and adults our primary hypothesis.

Some edits:This neuron, named Moonwalker/Mooncrawler Descending Neuron(MDN) is present in two bilateral pairs per brain lobe, – Awkward phrasing.

Thank you, corrected on page 2, lines 50-51. We now say “This neuron, named Mooncrawler/Moonwalker Descending Neuron (MDN) is present as a bilateral neuronal pairs in each brain lobe".

"Halting forward locomotion is done via activation of the Pair1 descendinginterneuron, which inhibits the A27h premotor neuron, to prevent it from inducing forward locomotion (Carreira-Rosario et al., 2018)." – Unclear.

Thank you, corrected on page 2, lines 54-57. We now say “Halting forward locomotion is achieved by via activation of the Pair1 descending interneuron, which inhibits the A27h premotor neuron. Given that the A27h interneuron is required for forward locomotion, its inhibition via MDN-induced Pair1 activation prevents forward locomotion".

"We find that all of these questions are answered in the affirmative, showing that the core MDN-Pair1 decision-making circuit (a pair of synaptically-connected interneurons) persists from larva to adult, despite profound remodeling during metamorphosis, and that this circuit coordinates forward/backward locomotion in both larvae and adults." – see point above."induce backward locomotion via the coordinate arrest of forward locomotion" – unclear.

Thank you, corrected on page 2, lines 53-54. We now say “Larval MDNs function within a neural circuit that induces backward locomotion and coordinately arrests forward locomotion".

77 – "but virtually all of the dendridic processes and descending axonal process had been pruned (Figure 1C, only one neuron labeled)."

Thank you, corrected on page 3, line 79. We now say “but virtually all of the dendridic processes and descending axonal process are pruned".

107 – "We conclude that the Pair1 neurons are present from larval to adult stages, and that the Pair1 neurons are enriched for postsynaptic partners in the T1neuromere." – "enriched" is a strange word in this context.

Thank you, corrected on Page 3, line 109. We now say "Pair1 neurons have many postsynaptic partners in the T1 neuromere."

[Editors' note: further revisions were suggested prior to acceptance, as described below.]

The reviewers have discussed their reviews with one another, and the Reviewing Editor has drafted this to help you prepare a revised submission.Essential revisions:Please shorten the Discussion and remove the Trans-Tango data as suggested by Reviewer #2.Reviewer #2:The authors have revised their manuscript, A locomotor neural circuit persists and functions similarly in larvae and adult *Drosophila*, which details the persistence of a circuit motif across metamorphosis, undergoing remodeling while maintaining a similar function. In my opinion, the writing has been significantly improved by raising and supporting the notion that remodeling on this scale has not been observed, both in the introduction and the discussion. The authors also found a positive, provocative way to suggest MDN-Pair1 participate in decision-making by citing recent results in *C. elegans*. The discussion is currently long and slightly repetitive with the intro and text; I would recommend another edit to shorten. For example, a paragraph dedicated to the identification of transcription factors in the discussion is unnecessary. Finally, the paper reads more clearly as an extension of Carreira-Rosario et al., 2018, which makes it more appropriate for an eLife Advance.I appreciate the addition of the Trans-Tango data, but they are of limited utility since downstream neurons are not clearly labelled or identifiable, and what labelling does exist could indicate downstream partners of the sensory neurons labeled by the Pair1 Gal4 line. I noticed in the results that the authors dissected the adult CNS 4 days post-eclosion. However, it generally takes much longer for the trans-tango signal to appear. The original paper (Talay et al., 2017) used 15-20 day old flies. I recommend that the authors' either repeat the experiments with a positive control to ensure that trans-tango is working, or remove them from the manuscript entirely.

We have precisely followed the reviewer’s suggestions in the text: (1) we remove the trans-tango data; (2) we shorten the discussion, including removing reference to trans-tango and deleting two entire paragraphs including the one reviewer suggested (on molecular markers).